# Phagocytic activity of blood monocytes and neutrophils in moderate COVID-19 patients and impact of immune therapy with bacterial lysates

Mikhail Kostinov[1,2], Oksana Svitich[1], Alexander Chuchalin[3], Viliya Gajnitdinova[2], Irina Bisheva[1], Svetlana Skhodova[1], Valerij Osiptsov[4], Vitalij Tatevosov[4], Nadezhda Kryukova[3], Isabella Khrapunova[2], Alexander Cherdantsev[5], Irina Soloveva[5], Nelli Akhmatova[1], Ekaterina Kurbatova[1], Valentina Polishchuk[1], Aristitsa Kostinova[2*], Anna Vlasenko[6], Marina Loktionova[2], Arseniy Poddubikov[2]

1 I. Mechnikov Research Institute of Vaccines and Sera, Moscow, Russian Federation, 2 I.M. Sechenov First Moscow State Medical University, Moscow, Russian Federation, 3 Pirogov Russian National Research Medical University, Moscow, Russian Federation, 4 The Main Military Clinical Hospital of the National Guard Troops, Moscow, Russian Federation, 5 Ulyanovsk State University, Ulyanovsk, Russian Federation, 6 Branch Campus of the Russian Medical Academy of Continuous Professional Education, Novokuznetsk State Institute for Advanced Training of Physicians, Novokuznetsk, Russian Federation

* aristica_kostino@mail.ru

## Abstract

### Background

Monocytes, macrophages and dendritic cells are involved in phagocytic reactions, which potentially play an important role in the pathogenesis of COVID-19. Imbalance of these cells in peripheral blood has proven to affect not only innate but also adaptive immunity. It is possible that a search for strategies to restore monocyte activity could be a major step in achieving immune control over COVID-19. The aim of this study was to investigate the relationship between phagocytic activity of peripheral-blood monocytes and neutrophils, and COVID-19 severity, to assess the effects of a bacteria-based immunostimulating agent on phagocytosis parameters in in-hospital COVID-19 patients.

### Materials and methods

The study included 105 adult patients with moderate COVID-19, who had been hospitalized in 2020–2021 and treated in accordance with the recommendations of the Ministry of Health of the Russian Federation. All patients were divided into two groups: in Group 1 patients received standard treatment and Immunovac VP4 therapeutic vaccine, a bacteria-based immunostimulating agent as add-on therapy from Day 1 of hospitalization; in Group 2 patients did not receive any add-on treatment. The study parameters included C-reactive protein (CRP), aspartate aminotransferase (AST), $SpO_2$, lung involvement on chest computer tomography (CT) scan, and phagocytic activity of peripheral-blood leukocytes based on the absorption activity

**Data availability statement:** All relevant data are within the manuscript.

**Funding:** The author(s) received no specific funding for this work.

**Competing interests:** The authors have declared that no competing interests exist.

(AA) of monocytes and neutrophilic granulocytes against *S. aureus)*. The parameters were assessed at 1, 14 and 30 days.

## Results

Based on a cluster analysis of the clinical findings and the results of diagnostic tests obtained on admission, the patients were divided into 2 clusters: cluster 1 including patients with a more severe disease (n = 34) and cluster 2 including patients with a less severe disease (n = 71). Cluster 1 patients had higher levels of CRP (20.1 *vs.* 2.2 mg/mL, p < 0.001), AST (32.9 *vs.* 26.2 U/L, p = 0.003), lower $SpO_2$ (94% *vs.* 96%, p < 0,001) and more extensive lung involvement on chest CT scan (35% *vs.* 12%, p < 0,001). There was a statistically significant direct correlation between blood monocyte AA and $SpO_2$ (p = 0.04), an inverse correlation between monocyte AA and CRP (p = 0.003) and the extent of lung involvement on CT scan (p = 0.05). In less severe COVID-19 patients (cluster 2), no statistically significant correlation was observed. In more severe COVID-19 patients (cluster 1), there was a rise in monocyte AA on day 30 of hospitalization both in the control group (from 86.6 to 92.2, p = 0.03) and the main group, who received Immunovac VP4 add-on therapy (from 87.3 to 98.3, p = 0.05). However, the patients who received the immunostimulating agent, had higher monocyte PI than the controls, without the immunostimulant (p = 0.05). Patients from cluster 1 who were given Immunovac VP4 had higher $SpO_2$ levels (98% *vs.* 97%, p = 0.01) than those who had received only the standard treatment.

## Discussion

Blood monocyte AA correlates with COVID-19 severity: patients with less severe disease have higher AA and those with more severe illness have lower AA. The standard treatment, combined with Immunovac VP4 enhances phagocytic activity of peripheral-blood monocytes, which is associated with a more marked increase in $SpO_2$, especially in more severe patients.

## Introduction

A number of studies have already demonstrated that among all immune cells it is phagocytes that are in the forefront of the host defense against viral infections and play a central role in generating immunity [1]. Phagocytosis can be followed by inflammatory pathway activation, which promotes pathogen elimination and inhibits pathogen growth [2]. Macrophages, neutrophils, monocytes, dendritic cells, and osteoclasts are termed professional phagocytes. Phagocytes express several receptors that activate signaling pathways resulting in phagocytosis [3,4].

Fibroblasts, epithelial cells, endothelial cells, and B-cells are described as non-professional phagocytes. They can also accomplish phagocytosis with low-efficiency and mainly internalize antigens through endocytosis. These cell are important in eliminating dead cells and maintaining homeostasis [5–8].

Findings obtained during the COVID-19 pandemic supported the hypothesis that there is a relationship between the occurrence of severe immune disorders in COVID-19 patients and infection of immune cells with SARS-CoV-2. Certain phagocytic cells, such as monocytes and macrophages, were found to contribute to the local tissue inflammation and cytokine storm in COVID-19 patients [9].

Monocytes and macrophages are not only engaged at all the stages of the inflammatory response but are also capable to regenerate tissues [10,11]. Patients with SARS-CoV-2 infection had higher peripheral-blood monocyte counts [12–14]. Analysis of circulating monocytes helps predict the severity and mortality associated with COVID-19 [15,16].

COVID-19 is often accompanied by secondary bacterial infection. The conducted study provides new insights into an efficient treatment for COVID-19 accompanied infectious with the use of bacterial lysate Immunovac VP4. In this study we evaluated the relationship between leukocyte phagocytic activity against *S. aureus* and clinical status on admission in patients hospitalized with moderate COVID-19. The patients were divided into two groups, which were provisionally termed as groups of more severe and less severe patients. This division was done based on the values of the following variables used in the cluster analysis: $SpO_2$ as an indicator of lung damage, C-reactive protein (CRP) as a marker of inflammation intensity, and aspartate aminotransferase (AST) as another indicator of degree of systemic inflammation in COVID-19. After that, were investigated these parameters in relation to the treatment regimen administered to the participants. For this purpose, the patients were divided into the following two groups: control group (41 patients), which received only the standard treatment, and main group (64 patients), which received the standard treatment in combination with Immunovac VP4 vaccine, a bacteria-based immunostimulating agent.

## Materials and methods

### Materials

**Study design.** It was a single-center non-interventional longitudinal observational study, which was conducted in a tertiary hospital for COVID-19 patients (Moscow) and the I.I. Mechnikov Research Institute of Vaccines and Sera (Moscow, Russia), between November 30, 2020 and May 30, 2021.The study was approved by the Ethics Committee at the I.I. Mechnikov Research Institute of Vaccines and Sera (Moscow) (Ethics Committee meeting №11/2020, dated November 26, 2020).

All patients received treatment in accordance with the Provisional Clinical Guidelines "Prevention, diagnosis, and treatment of novel coronavirus infection (COVID-19)" developed by the Ministry of Health of the Russian Federation, version 9, dated October 26, 2020. It included Favipiravir 200 mg (standard regimen), enoxaparin 0.4 mg/day, subcutaneously, dexamethasone 8–12 mg/day, and tocilizumab 400 mg/day (for patients with CRP ≥ 60 mg/L). The selection of patients was done following physical examination and screening assessments carried out on day 1 of hospitalization and was based on the inclusion and non-inclusion criteria.

During stage *one* of the study, a cluster analysis was performed using the results from laboratory and other diagnostic tests to divide patients into two groups, which were provisionally termed as groups of more severe and less severe disease at the time of admission. The patients' clinical condition at baseline was characterized using the following key parameters: CRP, AST, $SpO_2$ and percentage of lung involvement on chest computed tomography (CT) scan. Following this, phagocytic activity of peripheral-blood leukocytes was assessed by determining monocyte and neutrophil absorption activity (AA) against *S. aureus* and a correlation analysis was carried out to examine the relationship between the percentage (%) of phagocytic monocytes, $SpO_2$ and the percentage of lung involvement on chest CT.

In stage *two* of the study, phagocytic activity of blood monocytes and neutrophils was analyzed by treatment group: control group (n = 41), in which the patients received only the standard treatment, and main group (n = 64), in which the patients received the standard treatment in combination with Immunovac VP4 therapeutic vaccine, a bacteria-based immunostimulating agent. Immunovac VP4, which is included in the Russian National Registry of Medicinal Products, was administered in accordance with the information in the Indications and Contraindications sections of the Product Information Leaflet.

The patients were examined and their clinical status was assessed at baseline (on admission to hospital) and on days 14 and 30 of treatment. All treatment information, physical examination findings and study tests data were reported using standard medical records (individual patient documentation).

**Inclusion criteria.** Age 18–60 years, admission to hospital; CT signs of lung damage such as ground-glass opacities and areas of consolidation consistent with grade 2 of lung injury as assessed by CT scan (25–50% lung involvement); signed and dated informed consent.

**Exclusion criteria.** Patient's refusal to participate in the study; severe birth defects or severe chronic diseases; a history of cancer; a history of a positive human immunodeficiency virus (HIV) or hepatitis B or C tests; use of immunoglobulins or blood transfusion within the last three months prior to the start of the study; long use (more than 14 days) of immunosuppressive drugs or corticosteroids; any surgery within one month prior to inclusion in the study; autoimmune disease; simultaneous participation in another clinical study; or the patient's inability to comply with the study protocol requirements.

**Patients.** The study population was comprised of 69 males and 36 females. The mean age of the patients was 43.5 (37; 51) years old; the mean body mass index 27.4 (25; 30.1) kg/m2; and the mean duration of disease prior to hospitalization 6 (4; 8) days. According to the officially recommended severity criteria (T body > 38 °C; RR > 22/min, dyspnea during physical exertion; changes in CT (radiography) typical for viral infection; SpO2 < 95%; CRP in serum > 10 mg/l.) patients with moderate course of the disease were divided into two groups: less severe and more severe. This was done to see the dependence of changes in phagocytic activity on the severity of the disease. Since disease severity depends on a number of factors, a cluster analysis was carried out to classify all the patients (n = 105) into clusters based on several parameters (Table 1).

A comparison was also made between patients (n = 105) with moderate COVID-19 depending on the type of treatment received. The control group (group with standard treatment) consisted of 41 patients with moderate COVID-19 (27 males and 14 females, median age 42 [33–54]) who received only the standard treatment. The main group (group with standard treatment plus vaccine) was made up of 64 patients (42 males and 22 females, median age 42 [37–45]) who received the standard treatment accompanied by Immunovac VP4 vaccine, a bacteria-based immunostimulating agent, starting from day 1 of hospitalization.

The main and control groups were matched by age (p = 0.79), gender (p = 0.33) and the number of days between the onset of disease and hospitalization (5 [4–8] days in Group 1 and 5 [3–7] days in Group 2, p = 0,63). The patients in both groups were also matched by body mass index, amount of impaired lung parenchyma, and laboratory findings.

**Immunovac VP4 vaccine.** Immunovac VP4 (Certificate of Marketing Authorization No. ЛСР-001294/10 issued on February 24, 2010) belongs to a novel class of vaccines, so called "therapeutic vaccines". It is manufactured by Scientific and Production Association Microgen, a federal state unitary enterprise (Ufa, Russia). The vaccine exhibits protective activity against the microorganisms included in its composition and provides an immunomodulatory effect [17].

"Immunovac VP4" belongs to the class of therapeutic vaccines of bacterial origin. Analogues of "Immunovac VP4" are Bronchomunal, Bronchovaxom and Respivax, consisting of lysates of 7 types of opportunistic bacteria, intended for oral use only [18–20]. Immunovac VP4 has a number of advantages: it contains antigens of 4 types of specially selected highly immunogenic bacterial strains with broad intraspecific and interspecific cross-protective activity, including against various serotypes of *Streptococcus pneumoniae* and *Hemophilus influenzae*. The course of therapy with Immunovac VP4, administered orally or subcutaneously, is 1 month, the duration of the therapeutic effect is a year or more, while the course of treatment with Broncho-munal and its analogues is 3 months with the possibility of repeat after 6 months. Immunovac VP4 includes the antigens of *Staphylococcus aureus, Klebsiella pneumoniae, Proteus vulgaris* and *Escherichia coli* (4 mg). Using non-aggressive methods for treatment of microbial biomass made it possible to mostly maintain the native structure of the antigens and include protein-associated lipopolysaccharides isolated from *K. pneumoniae, P. vulgaris* and *E. coli*, proteoglycan isolated from *K. pneumoniae*, as well as teichoic acids derived

**Table 1. Clinical and laboratory characteristics of the study patients with moderate COVID-19 and characteristics of the identified clusters.**

| Parameter | All patients n=105 | Cluster 1 More severe disease n=34 (32.4%) | Cluster 2 Less severe disease n=71 (67.6%) | p value |
|---|---|---|---|---|
| | Me (SD)/ Med (Q1; Q3) | Me (SD)/ Med (Q1; Q3) | Me (SD)/ Med (Q1; Q3) | |
| Age, years | 43.5 (37; 51) | 44.8 (39; 51) | 42.9 (37; 49) | p=0.44 |
| Duration of disease prior to hospitalization, days | 6 (4; 8) | 6 (3.5; 7.5) | 5 (4; 8) | p=0.83 |
| Duration of fever prior to hospitalization, days | 1 (0; 4) | 1 (1; 5) | 1 (0; 4) | p=0.22 |
| RR, breaths/min | 18 (18; 20) | 21 (20; 21) | 18 (18; 18) | p<0.001 |
| HR, bpm | 74 (67; 87) | 90 (83.5; 94) | 69.5 (65.2; 76.8) | p<0.001 |
| SpO$_2$, % | 96 (95; 96) | 94 (94; 95) | 96 (96; 96) | p<0.001 |
| Lung involvement (on chest CT scan), % | 20 (14; 40) | 35 (30; 43.5) | 12 (7; 18) | p<0.001 |
| WBC, 10$^9$/L | 5 (3.7; 6.9) | 5.3 (3.6; 7.1) | 5 (3.8; 6.3) | p=0.84 |
| RBC, 10$^9$/L | 5 (0.6) | 4.8 (0.7) | 5 (0.5) | p=0.07 |
| Hemoglobin, g/L | 146 (135; 156) | 145 (136; 154) | 147 (134; 156) | p=0.93 |
| Platelets, 10$^9$/L | 221.8 (64.5) | 205 (53.6) | 228.6 (67.5) | p=0.06 |
| Segmented neutrophils, % | 66 (57.3; 75) | 69 (63; 77.5) | 64 (53.6; 72.2) | p=0.02 |
| Band neutrophils, % | 6.7 (3.8) | 6 (3.7) | 7.1 (3.9) | p=0.33 |
| Neutrophils, 10$^9$/L | 3.1 (2.2; 4.9) | 3.4 (2.5; 5.4) | 2.9 (2.1; 4.4) | p=0.30 |
| Lymphocytes, % | 23.5 (12.1) | 18.1 (8.6) | 25.7 (12.7) | p<0.001 |
| Lymphocytes, 10$^9$/L | 1.2 (0.7) | 0.99 (0.57) | 1.3 (0.7) | p=0.011 |
| ESR, mm/h | 25 (16; 34) | 28 (20; 38) | 21 (16; 30) | p=0.05 |
| Neut/Lymph ratio | 0.72 (0.31; 2.34) | 1.11 (0.19; 3.6) | 0.72 (0.37; 1.8) | p=0.94 |
| ALT, U/L | 30.9 (19.6; 48) | 35 (26; 50.5) | 29.1 (18.7; 44.7) | p=0.15 |
| AST, U/L | 29 (21; 39) | 32.9 (28.2; 44.9) | 26.2 (20; 34.9) | p=0.003 |
| CRP, mg/L | 5.5 (0.7; 20.1) | 20.1 (10.9; 43.2) | 2.15 (0.34; 10.67) | p<0.001 |
| D-dimer, ng/mL | 276 (104.5; 456) | 191 (100; 446.8) | 285 (166; 461) | p=0.46 |
| Fibrinogen, g/L | 3 (2.5; 3.8) | 3 (2.7; 4.2) | 2.9 (2.5; 3.7) | p=0.40 |
| % of phagocytic monocytes | 91.2 (84.6; 95) | 86.7 (81.4; 91.7) | 92.6 (86.5; 95.4) | p=0.012 |
| % of phagocytic neutrophils | 97.9 (96.3; 99) | 98 (95.2; 98.8) | 97.8 (96.5; 99) | p=0.73 |

RR=respiratory rate, HR=heart rate, SpO$_2$=oxygen saturation by pulse oximetry, AST=aspartate aminotransferase, ALT=alanine aminotransferase, CRP=C-reactive protein.

from *S. aureus* and protein and other polysaccharide antigens of S. aureus. The spectrum of antimicrobial activity of Immunovac VP4 against various opportunistic bacteria, including *S. pneumoniae* and *H. influenzae*, was expanded not by increasing the number of its components but by incorporating the antigens isolated from specifically selected highly immunogenic strains with weak sensitizing properties, which had intra- and inter-species protective antigens as well as immunostimulatory antigens. This considerably simplified the manufacturing process for this vaccine and increased its therapeutic effectiveness. Immunovac VP4 does not contain any preservatives or stabilizing additives. It is available as a lyophilized powder in ampoules and vials (see Product Information Leaflet for Immunovac VP4, a multi-component vaccine based on antigens of opportunistic microorganisms, approved by the Ministry of Health of the Russian Federation on September 18, 1996).

**Pharmacological properties.** It is a bacteria-based immunostimulant. Its mechanism of action is due to the activation of the key effectors of innate and adaptive immunity. This vaccine enhances phagocytic activity of macrophages, optimizes cell counts and functional activity of lymphocyte subsets (CD3, CD8, CD16, and CD72),

programs CD4 T-cells to proliferate and differentiate into Th1 cells, stimulates the production of interferon (IFN)-γ and IFN-α, and improves the production of immunoglobulin isotypes by inhibiting IgE synthesis and inducing IgG, IgA, and sIgA synthesis [21–23].

It induces the production of antibodies to four opportunistic microorganisms whose antigens are included in the composition. It also contains antigens that can induce broad cross-protection against other pathogens (*S. pneumoniae, H. influenzae* and others).

On the clinical side, Immunovac VP4 reduces the rate of acute infections, duration of acute infection, severity of symptoms, risk of exacerbation of chronic diseases, and the number and doses of medications used by patients.

**Indications for immunovac VP4.** Children older than 15 years of age and adults: chronic recurrent inflammatory respiratory diseases (acute phase [5–7 days after the start of standard treatment], remission, and periods of increased number of respiratory infection cases prior to respiratory infection seasons); allergic disorders, including mixed-type asthma, infection- and allergy-induced asthma, and atopic dermatitis (in combination with standard treatment during remission or after exacerbation); prevention of respiratory infections in individuals who have frequent acute respiratory disease evets (more than 4 events a year) (periods of increased number of cases prior to respiratory infection seasons).

Preparing of Immunovac VP4 solution: Immediately prior to use, 4 mL of solvent (0.9% sodium chloride for injection or boiled water brought to 18–25°C) is added to the vial with a syringe, and the contents is mixed. The dilution time should not exceed 2 minutes. The ready solution can be stored at +2–8°C for 3 days and can be used if it does not show any signs of cloudiness.

In our study, the following regimens, doses and schedules were used to immunize patients with moderate COVID-19: Regimen 1 (nasal and subcutaneous administration) (n = 31): When prepared, the vaccine solution was administered at a dose of 2 drops (1.0 mg) in each nostril daily and then subcutaneously every other day in the following amounts: 0.05 ml (0.5 mg) on day 1 of hospital stay, 0.1 ml (1.0 mg) on day 3, 0.2 ml (2.0 mg) on day 5, 0.2 ml (2.0 mg) on day 7, 0.3 ml (3.0 mg) on day 9, and 0.3 ml (3.0 mg) on day 11.

Regimen 2 (nasal and oral administration) (n = 33): When prepared, the vaccine solution was administered orally at a dose of 2 ml (20.0 mg) followed by 2 drops (1.0 mg) in each nostril daily from day 1 to day 10 of hospital stay.

Although the regimens used in each group were different, the study groups were joined in one group considering similar dynamics of sIgA production in the upper airways and similar clinical effects of an immunotherapeutic agent in COVID-19 patients, shown by other studies [24,25].

## Methods

The following parameters were assessed in all patients: demographic characteristics, duration of disease prior to hospitalization, treatment given prior to hospitalization, physical examination findings, laboratory test results obtained on admission (complete blood count, blood chemistry, CRP, blood coagulation profile, D-dimer, and leukocyte phagocytic activity), and chest CT findings.

A complete blood count test (red blood cell count, white blood cell count, absolute and relative counts of neutrophils, lymphocytes, monocytes, eosinophils, and basophils) was carried out on an XE 2100 automated hematology blood analyzer (Sysmex Corporation, Japan). This test was done on whole blood samples collected on admission.

Phagocytic activity of monocytes and granulocytes (neutrophils) was assessed in peripheral blood samples by measuring leukocyte absorption activity against heat-killed *S. aureus* labeled with fluorescein isothiocyanate. Blood samples were analyzed by flow cytofluorometry on an FC-500 BeckmanCoulter flow cytometer. Daily cultures, the second passage of S. aureus Wood 46, were washed with isotonic sodium chloride solution, killed by heating to 96−98 ° C for 40 minutes, precipitated at 1000 g for 25 minutes, washed twice in 10 ml of phosphate-buffered saline (PBS), pH 7.4. According to the turbidity standard, the concentration of bacteria was brought to 200 million/ml with

carbonate-bicarbonate buffer pH 9.5. FITC (Sigma) was added to the bacterial suspension at a final concentration of 0.1 mg/ml and incubated at +4°C for 12 hours. Then unbound FITC was removed by washing three times PBS solution on 1000 g for 25 minutes. The bacterial suspension was aliquoted and stored at +4°C for 1 month, at −70°C up to 6 months. Whole heparinized blood was used to set up the reaction. A suspension of FITC-labeled staphylococci and blood cells in a ratio of 1:10 was placed in Eppendorf tubes and incubated at 37°C for 30 minutes. Quenching of FITC-labeled bacteria adhered to the surface of leukocytes was performed by adding a solution of trypan blue (0.2 mg/ml). Then Optilyse C solution was added to lyse erythrocytes and the samples were incubated at room temperature in the dark for 30 minutes. Next, cold phosphate-buffered saline, pH 7.2–7.4, with 0.02% EDTA was added to stop the phagocytic reaction. After 3-fold washing with ISOTON II solution, the samples were analyzed on a Beckman Coulter FC-500 flow laser cytometer. The cytometer settings were set so that three cell clouds – granulocytes (neutrophils), monocytes, and lymphocytes – were conveniently placed on the forward scatter (FSC) and side scatter (SSC) diagram in the Dot Plot window. Phagocytic cells were counted based on the intensity of green fluorescence (FL-1 channel). The percentage of phagocytic granulocytes and monocytes in the test sample was based on the cytometry results. The recommended number of events collected for neutrophils is 3000.

Blood samples were processed by the Laboratory for Vaccination and Immunotherapy of Allergic Diseases at the Federal State Budgetary Scientific Institution I.I. Mechnikov Research Institute of Vaccines and Sera. This was done using certified equipment provided by the Institute's Research Equipment Sharing Center.

Lung computed tomography was performed on admission on a spiral CT scanner Aquilion TSX-101A (Toshiba Medical Systems, Japan, slice thickness 1 mm, pitch 1.5). The intensity (volume, area, and extent) of lung involvement in patients with suspected/known COVID-19-associated pneumonia was assessed using an adapted "empirical" visual assessment scale, which makes it possible to assess the approximate volume of lung opacities in the most severely injured lung regions: no characteristic signs (CT-0); minimal volume/extent (<25% of lung parenchyma) (CT-1); moderate volume/extent (25−50% of lung parenchyma) (CT-2); considerable volume/extent (50−75% of lung parenchyma) (CT-3); subtotal volume/extent (>75% of lung parenchyma) (CT-4).

**Statistical analyses.** Descriptive statistics included median and interquartile range for non-normally distributed quantitative variables and mean and standard deviation for normally distributed quantitative variables. The normality of distribution was tested using the Shapiro Wilk's normality test. Comparison of quantitative variables in two independent groups was done using the Student's test (for normally distributed variables) and the Mann—Whitney's test (for non-normally distributed variables).

The cluster analysis was done using the k-means method with the silhouette score applied to determine the quality of clusters. The correlation analysis was performed using the Spearman's rank correlation coefficient, and the robust linear regression analysis was carried out to calculate a regression line and construct a 95% confidence interval (CI) for its slope.

Linear mixed-effects models (LMEM) were used to evaluate the relationship between the percentage of phagocytic leukocytes (granulocytes and monocytes) and the severity of a patient's condition on admission, where time, severity of condition and their interaction (time × severity) were fixed effects, and patients were random effects. These models were estimated using in the lme4 package. Each model's fit (residual normality and homoscedasticity) was checked using the DHARMA package. When the model fit was poor, another LMEM was fit using the robust estimation method in the robustlmm package (robLMEM). Satterthwaite's approximation was used to calculate the degrees of freedom for the resulting models. Marginal R-squared ($R^2m$, only fixed effects) and conditional R-squared ($R^2C$, fixed and random effects) were generated for each model. Post-hoc tests were performed by estimating the corresponding contrasts in the LMEM using the lmerTest packages [26].

The level of statistically significant differences was defined as $p \leq 0.05$. The calculations and graphics were carried out using the statistical programming environment R (v.3.6, license GNU GPL2).

## Results

### Stage 1

The cluster analysis was carried out using the following factors: $SpO_2$ as an indicator of lung damage, CRP as a marker of inflammation intensity, and AST as another indicator of degree of systemic inflammation in COVID-19. Fig 1 shows the results of the cluster analysis. The silhouette score, a characteristic of the quality of clusters, was 0.65, which indicated that our clusters were well-defined [27,28].

Compared to cluster 2, cluster 1 had higher levels of CRP (20.1 [10.9; 43.2] mg/L vs. 2.2 [0.3; 10.7] mg/L, p < 0.001) and AST (32.9 [28.2; 44.9] U/L vs. 26.2 [20; 34.9] U/L, p = 0.003) and lower $SpO_2$ levels (94 [94; 95]% vs. 96 [96; 96]%, p < 0.001) (Fig 2). The percentage of lung involvement on CT scan was not included in the cluster analysis because of its strong correlation with $SpO_2$ (which, in this case, had a slightly higher differentiating ability). However, there was a significant difference between the clusters in the median percentage of lung involvement (35 [30; 43.5]% vs. 12 [7; 18]%, p < 0.001) (Fig 2).

Thus, the patients in cluster 1 (n = 34) were regarded as more severe, and patients in cluster 2 (n = 71) as less severe. According to the Russian clinical guidelines [28] such clinical findings are identified as signs of moderate COVID-19. There was no correlation between disease severity, on the one hand, and disease duration prior to hospitalization (6 [3.5; 7.5] days in the group of more severe patients and 5 [4; 8] days in the group of less severe patients, p = 0.83) or patient age (45 [38; 51] years old and 43 [37.2; 47] years old, respectively, p = 0.47), on the other hand. The group of more severe patients had a slightly higher proportion of males than the group of less severe patients (80% [25/31] vs. 60% [45/74], p = 0.08). The duration of fever prior to hospitalization was similar in the study groups (1 [1; 5] day in the group of more severe patients и 1 [0; 4] day in the group of less severe patients, p = 0.22).

Phagocytic activity of peripheral-blood leukocytes was assessed by the percentage (%) of phagocytic monocytes and phagocytic neutrophils against *S. aureus*. The median neutrophil PI was 97.9 (96.3; 99)%, and the median monocyte PI was 91.2 (84.6; 95)%.

The correlation analysis revealed a significant direct relationship between % of phagocytic monocytes and $SpO_2$ (ρ = 0.21 [95% CI: 0.005 to 0,39], p = 0.04) and an inverse relationship between % of phagocytic monocytes and CRP (ρ = −0.31 [95%

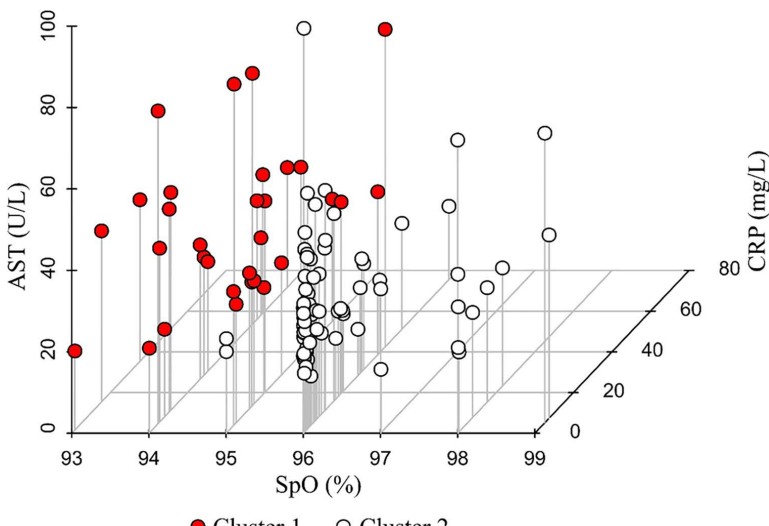

**Fig 1. The results of the cluster analysis of the data collected from moderate COVID-19 patients on admission to hospital.**

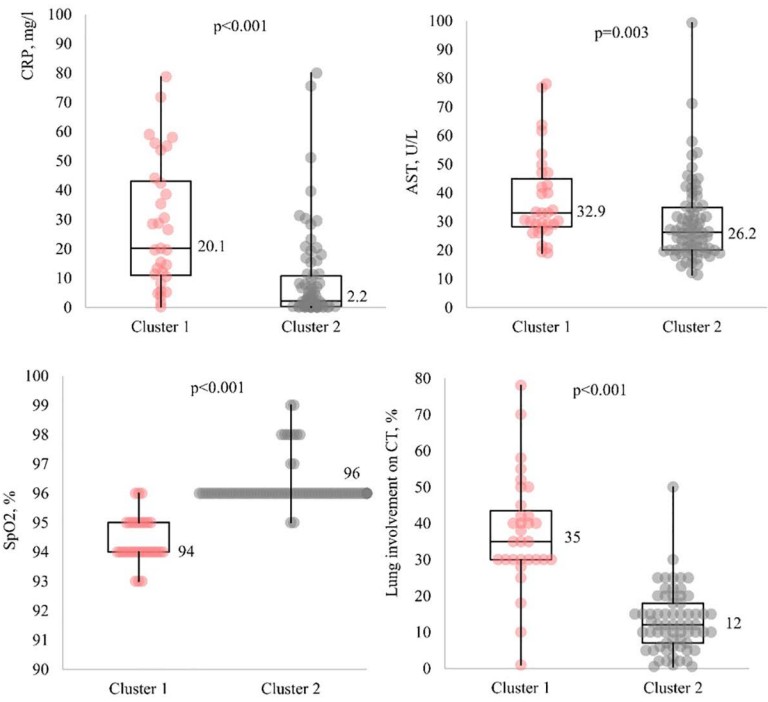

**Fig 2. The results of the cluster analysis of the data collected from moderate COVID-19 patients on admission to hospital (individual effects).**

CI: −0.11 to −0.48], p = 0.003) and between % of phagocytic monocytes and the percentage of lung involvement on CT scan (ρ = −0.2 [95% CI: −0.11 to −0.48], p = 0.05), Fig 3. Thus, higher % of phagocytic monocytes on admission was associated with lower CRP levels, lower percentages of lung involvement on CT scan and higher SpO$_2$ levels.

**Stage 2**

During the next phase, phagocytic activity of blood monocytes and neutrophils were analyzed by treatment group (the control group received the standard treatment, and the main group received the standard treatment in combination with Immunovac VP4) separately for more and less severe patients. Table 2 describes the dynamic models for % of phagocytic monocytes and neutrophils (robust linear mixed-effects models).

In the less severe patients, absorption activity of monocytes against *S. aureus* did not show a significant correlation with either measurement time point (baseline and days 14 and 30, respectively) or treatment group and was 93 (84.9–95,5), 91.8 (86.0–94,6), and 89.2 (81.7–95,2) in the control group, and 92.3 (87.1–95.3), 93.25 (91.15–95.9), and 90.3 (85.8–93.4) in the Immunovac VP4 group.

Similarly, the absorption activity of neutrophils did not differ between the study groups. In the less severe patients, % of phagocytic neutrophils at baseline and on days 14 and 30 was 97.7 (95.6–99), 98.55 (96.4–99.5), and 98.9 (97.6–99.6) in the control group and 97.9 (96.6–99), 98.4 (97.2–99.3), 99 (97.7–99.5) in the Immunovac VP4 group, respectively. In the more severe patients, % of phagocytic neutrophils at baseline and on days 14 and 30 was 97.2 (95.07–98,9), 97.5 (96.65–98,75), and 98.9 (97.8–99,65) in the control group, and 98.3 (95.7–98.7), 99.2 (98.5–99.7), and 96.1 (92.9–97.4) in the Immunovac VP4 group.

In the cluster of patients with a more severe disease at baseline, there was a significant rise in % of phagocytic monocytes from baseline to day 30 of treatment (p = 0.02), but the extent of these changes depended on the treatment regimen

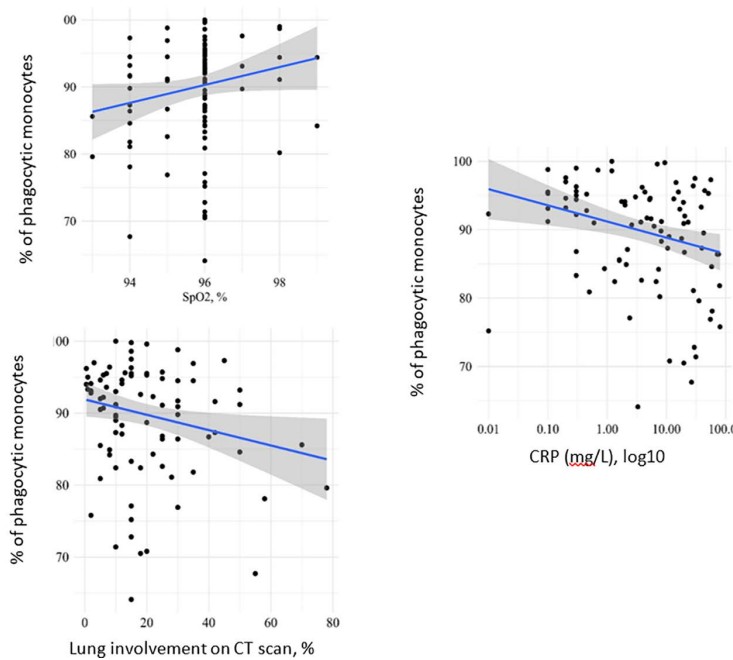

**Fig 3. The relationship between % phagocytic monocytes and SpO2, CRP and percentages of lung involvement on CT scan (on admission) in patients with moderate COVID-19.** The regression line was calculated and a 95% confidence interval for its slope was constructed by robust linear regression analysis.

**Table 2. Characteristics of the dynamic models for % of phagocytic monocytes and neutrophils by disease severity at baseline and treatment group (robust linear mixed-effects models were used).**

| Factors | % of phagocytic monocytes | | % of phagocytic neutrophils | |
|---|---|---|---|---|
| | Less severe | More severe | Less severe | More severe |
| Day 14 | p=0.70 | p=0.45 | p=0.22 | p=0.54 |
| Day 30 | p=0.17 | **p=0.02** | p=0.29 | p=0.18 |
| Treatment group | p=0.95 | p=0.58 | p=0.56 | p=0.72 |
| Day 14 time point×group | p=0.40 | p=0.78 | p=0.76 | p=0.63 |
| Day 30 time point×group | p=0.74 | **p=0.05** | p=0.68 | p=0.21 |
| Model quality | Rm=0.04 Rc=0.04 | Rm=0.28 Rc=0.28 | Rm=0.05 Rc=0.05 | Rm=0.08 Rc=0.10 |

Rm – marginal R-squared (only fixed effects), Rc – conditional R-squared (fixed and random effects).

(p=0.05) (Fig 4): % of phagocytic monocytes increased from 86.6 (81.2–91.6) to 92.2 (87.85–96.7) (p=0.03) in the control group, and from 87.3 (81.45–92.05) to 98.3 (97.3–99.6) (p=0.05) in the Immunovac VP4 group, and the difference between the groups was statistically significant (p=0.05).

In the less severe patients, there was a significant increase in $SpO_2$ from baseline to day 14 of treatment (p<0.001), and the magnitude of the increase was similar in the controls and the patients who received the immunostimulating agent. In the more severe patients, an increase in $SpO_2$ was observed in both groups (p<0.001), but it was more significant in the Immunovac VP4 group: baseline $SpO_2$ levels were slightly lower in the Immunovac VP4 group compared to the control

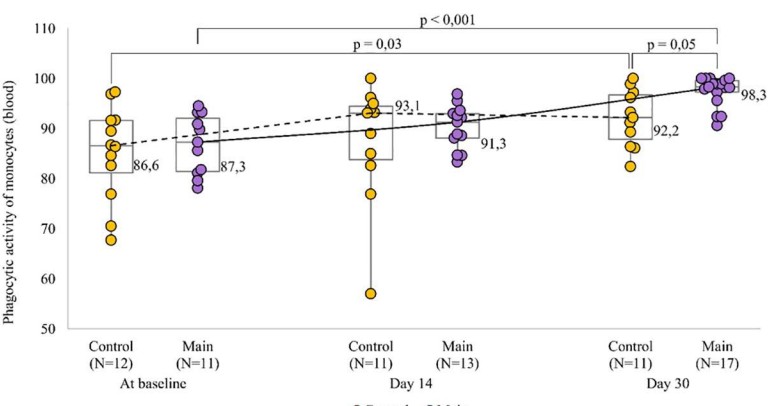

**Fig 4. The dynamics of phagocytic activity of monocytes in moderate COVID-19 patients with a more severe disease at baseline depending on treatment regimen.**

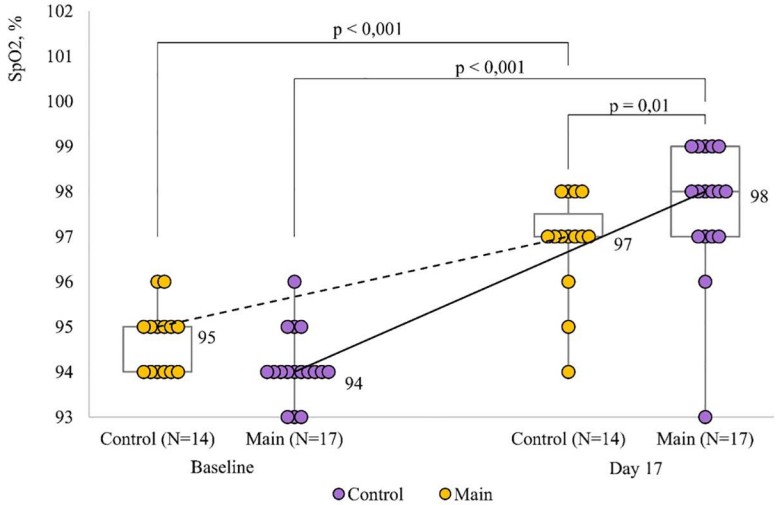

**Fig 5. The dynamics of oxygen saturation by pulse oximetry (SpO2) in moderate COVID-19 patients with a more severe disease at baseline depending on treatment regimen.**

group (94 [94; 94)% *vs.* 95 [94; 95]%, respectively, p = 0.08), but on day 14 of treatment they were higher than in the control group (98 (97; 99)% *vs.* 97 (97; 97.8)%, p = 0 .01) (Fig 5).

Another study parameter was the duration of fever in days. In the patients with more severe COVID-19, fever persisted for 6.5 (95% CI: 4 to NA) days in the control group and for 4 (95% CI: 1–7) in the Immunovac VP4 group (p=0.08) In the cluster of patients with less severe COVID-19, there was a significant correlation between the duration of fever and the treatment regimen: (5 [95% CI: 4–7] days in the control group *vs.* 3 [95% CI: 2–3] days in the Immunovac VP4 group, p=0.001) (Fig 6).

The duration of hospitalization (number of days in hospital) did not correlate with the treatment regimen and was 28.5 (95% CI: 25–34) days in the control group and 26 (95% CI: 23–35) days in the Immunovac VP4 group, p=0.39). However, it should be noted that in the cluster of more severe patients the duration of hospitalization was significantly higher than in the cluster of less severe patients: 22 (17.5; 26.5) days *vs.* 17 (14; 21.8) days, p=0.01.

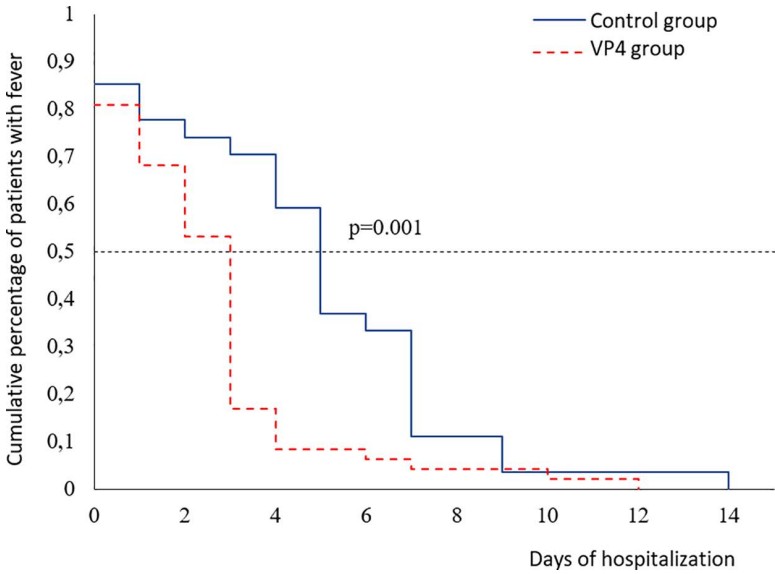

**Fig 6. Kaplan-Meier curves for the duration of fever in patients with less severe pneumonia by treatment regimen.**

## Discussion

In the early stages of the disease, innate immunity plays an important role in defense against viral infections and helps prevent viremia. Moreover, phagocytic activity of macrophages, monocytes, dendritic cells, and neutrophils is essential for triggering adaptive immunity. These cells detect viral infection through pattern-recognition receptors (PRRs), which recognize pathogen-associated molecular patterns (PAMPs) and damage-associated molecular patterns (DAMPs) [29].

In COVID-19, the activation of resident alveolar macrophages, CD68+CD169+ macrophages of lymph nodes, and CCR2+ monocytes through PRRs triggers the production of type I IFN-γ and other pro-inflammatory cytokines, such as interleukin (IL)-6, IL-1β, and tumor necrosis factor (TNF)-α, that have effective antiviral functions, but also contribute to lung tissue damage and inflammation [30]. Macrophages and monocytes produce high levels of chemokines (CCL2, CXCL8, CXCL10, etc.) that can recruit other innate and adaptive immune cells to the site of infection [31]. Therefore, phagocytes coordinate the inflammatory response and play a pivotal role in the interaction between innate and adaptive immunity [30,32].

In severe COVID-19 an overproduction of proinflammatory cytokines and chemokines results in the enhancement of neutrophil activity associated with immune pathology. Neutrophils are another key component of innate immunity. They also have a distinct array of other immune roles, such as NETosis (liberation of neutrophil extracellular traps) for viral infection inactivation and cytokine production to restrict virus [30]. In viral infections, neutrophils migrate to the site of pathogen invasion and contribute to the killing of viruses through active oxygen series (ROS) and phagocytosis. Thus, neutrophils can initiate, amplify and/or suppress adaptive immune effector responses, contributing to bidirectional crosstalk with T-cells [33,34].

Several studies revealed that the presence of neutrophils and macrophage cluster-1 is a hallmark of severe COVID-19. Further analyses revealed genes associated with the inflammatory response and chemotaxis of myeloid cells, phagocytes, and granulocytes, in bronchoalveolar lavage from severe COVID-19-affected patients [35].

There is a growing body of evidence that phagocyte dysfunctions in patients with COVID-19 and lung damage are linked with an increased susceptibility to secondary infections [36–43]. Several studies found that patients with COVID-19 may also develop secondary bacterial or fungal infections. An upregulation of activation markers and downregulation of *S. aureus* phagocytosis in circulating phagocytes in COVID-19 patients was demonstrated by Koenis D.S. et al. [44].

Considering a wide range of the studies investigating phagocytosis in COVID-19, we chose to conduct a study to evaluate the absorbtion activity of blood monocytes and neutrophils against *S. aureus* in patients with moderate COVID-19 and to assess the efficacy of an immunotherapeutic agent administered in combination with the standard treatment recommended for this condition. The phagocytic activity of blood monocytes and neutrophils was assessed before treatment and at several time points during treatment in a group of patients who received the standard treatment recommended by the national guidelines and in a group of patients who were given Immunovac VP4, a bacteria-based immunostimulating agent, as part of a combination treatment regimen. The process of phagocytosis involves several phases: 1) detection of the particle to be ingested, 2) activation of the internalization process, 3) formation of a specialized vacuole called phagosome, and 4) maturation of the phagosome to transform it into a phagolysosome. Our study focused, in fact, on phases 3 and 4 of phagocytosis and assessed the phagocytic activity of peripheral-blood leukocytes against *S. aureus.* as well as the percentage of fluorescent (phagocytizing) neutrophils and monocytes (neutrophil and monocyte PIs).

Before treatment, the median absorption activity of monocytes and neutrophils against *S. aureus* were within the reference ranges in this population of patients hospitalized for moderate COVID-19. The correlation analysis revealed a significant direct correlation between phagocytic activity of blood monocytes and oxygen saturation by pulse oximetry and an inverse correlation between phagocytic activity of blood monocytesand CRP levels and the percentage of lung involvement on CT scan. Thus, higher absorption activity of blood monocytes on admission was associated with lower CRP levels, lower percentages of lung involvement on CT scan and higher oxygen saturation. The phagocytic activity of monocytes was lower in more severe patients. Other authors also reported a reduction in the percentage of peripheral-blood monocytes in patients with severe COVID-19 [45–47].

In our study, however, no trends or differences in blood neutrophils absorption activity were observed either on admission or during treatment between the group receiving only the standard treatment and the group receiving the standard treatment in combination with a bacteria-based immunostimulating agent (Immunovac VP4).

Similarly, no significant relationship was observed between blood monocyte phagocytic activity against *S. aureus*, on the one hand, and measurement time point or treatment group, on the other hand, in the less severe patients. Nonetheless, in the cluster of patients with a more severe disease at baseline, there was a significant rise in % of phagocytic monocytes from baseline to day 30 of treatment (p = 0.02), but the extent of these changes depended on the treatment regimen (p = 0.05). The patients who received Immunovac VP4 in combination with the standard treatment had higher % of phagocytic monocytes than the patients who did not receive the immunotherapeutic agent. Previous studies showed that Immunovac VP4 combined with the standard treatment enhanced the phagocytic activity of peripheral-blood monocytes in asthma patients and those with pyoderma [48].

In addition, Immunovac VP4 induces maturation of dendritic cells [49].

Dendritic cells activated by Immunovac VP4 produce IL-12, which is the leading cytokine that programs T-cells to differentiate into Th1 cells, which together with other cytokines play a key role in generating defense against various pathogens [50–53].

Our study demonstrated that the standard treatment alone and in combination with Immunovac VP4 resulted in recovery from COVID-19. However, it cannot be excluded that treatment with an immunotherapeutic agent stimulated resolution of inflammation. A number of previous studies showed that patients with respiratory infections who received an immunostimulating agent (Immunovac VP4) made a quicker recovery than those who received only standard treatment [54].

In our study, both groups of COVID-19 patients showed a significant increase in $SpO_2$ (p < 0.001) on day 14 of hospital stay. However, the more severe patients who received Immunovac VP4 had higher levels of oxygen saturation (p = 0.01), which suggests clinical benefits of immunotherapy. Moreover, in the patients with less severe COVID-19 who received Immunovac VP4 the mean duration of fever was 2 days shorter than in those who received only the standard treatment (p = 0.001).

This study had several limitations. We did not assess the same parameters in patients with severe and very severe COVID-19 because of concerns about the possibility of side effects of immunotherapeutic treatment, which was explained by the absence at that time of any international experience in using immunostimulating agents as part of combination treatment for COVID-19 patients.

We hope that our results provide further information that could be useful in identifying potential therapeutic interventions based on immune changes in COVID-19.

## Conclusion

Absorbtion activity of blood monocyte correlates with COVID-19 severity on admission to hospital: high levels are associated with a lower severity of disease, and low levels are associated with a more severe disease. The standard treatment, combined with Immunovac VP4, an immunostimulating agent, enhances phagocytic activity of peripheral-blood monocytes and provides a more pronounced increase in SpO2, especially in more severe patients.

## Summary

In COVID-19 patients, the phagocytic activity of monocytes was inversely related to the severity of disease on admission to hospital: patients with less severe COVID-19 had higher % of phagocytic blood monocyte. Among the patients with more severe COVID-19, those who received a bacteria-based immunostimulating agent (Immunovac VP4 vaccine) and the standard treatment from day 1 of hospitalization had higher % of phagocytic monocytes. The COVID-19 patients who received immunotherapeutic treatment had higher oxygen saturation levels and a shorter duration of fever than the patients who did not receive an immunostimulant as part of combination therapy.

## Author contributions

**Conceptualization:** Mikhail Kostinov.

**Data curation**: Viliya Gajnitdinova, Valerij Osiptsov, Vitalij Tatevosov, Nadezhda Kryukova, Isabella Khrapunova.

**Formal analysis:** Mikhail Kostinov.

**Investigation:** Irina Bisheva, Svetlana Skhodova, Nelli Akhmatova, Arseniy Poddubikov.

**Methodology:** Oksana Svitich.

**Project administration:** Alexander Chuchalin.

**Software:** Viliya Gajnitdinova.

**Supervision:** Marina Loktionova.

**Validation:** Alexander Cherdantsev, Irina Soloveva.

**Visualization:** Ekaterina Kurbatova, Valentina Polishchuk, Anna Vlasenko.

**Writing – original draft:** Mikhail Kostinov.

**Writing – review & editing:** Nelli Akhmatova, Aristitsa Kostinova.

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
