## [Decision Letter · Decision Letter 0]

19 Feb 2025

PLOS ONE

Dear Dr. Kostinov,

We look forward to receiving your revised manuscript.

Kind regards,

Mrinmoy Sanyal, PhD

Academic Editor

PLOS ONE

Reviewers' comments:

Reviewer's Responses to Questions

**Comments to the Author**

1. Is the manuscript technically sound, and do the data support the conclusions?

Reviewer #1: Yes

Reviewer #2: No

Reviewer #3: No

Reviewer #4: Partly

2. Has the statistical analysis been performed appropriately and rigorously?

Reviewer #1: Yes

Reviewer #2: No

Reviewer #3: Yes

Reviewer #4: Yes

3. Have the authors made all data underlying the findings in their manuscript fully available?

Reviewer #1: Yes

Reviewer #2: Yes

Reviewer #3: No

Reviewer #4: Yes

4. Is the manuscript presented in an intelligible fashion and written in standard English?

Reviewer #1: Yes

Reviewer #2: No

Reviewer #3: Yes

Reviewer #4: No

Reviewer #1: Very Respected Authors,

COVID-19 is still relevant. Your paper is well written. The abstract is well organized. The aim is clear. The methodology is well explained, and you have obtained approval from the ethics committee. The appropriate statistical analysis was chosen. The results are clearly presented, and the discussion effectively explains the findings in relation to other research results. Relevant references have been selected.

Reviewer #2: The manuscript by Mikhail Kostinov, et. al., describes a study of phagocytosis by monocytes and neutrophils from patients with COVID-19. Authors indicate that phagocytosis in deficient in these cells and it is restored after treatment and improvement of patients' health conditions. Also, they compare the standard treatment with the standard treatment plus a bacterial antigen vaccine, called Immunovac VP4.

Previous reports have suggested that cells of the innate immune system, particularly phagocytic cells, have a role in the pathogenesis of the disease. Hence, exploring the function of these cells is relevant. However, the present report does not properly address the phagocytic functions of cells. The manuscript is written in a confusing manner because many concepts are not described appropriately, particularly at the beginning of the manuscript. For example, "bacteria-based immunostimulating agent, CRP, AST, SpO2, and Immunovac VP4" are not defined until very late in the manuscript. So, the reader cannot follow what the authors are referring to. Also, it is not clear what the control group is and what an "immunostimulating agent" is. Moreover, it is stated that "control group (41 patients), which received only the standard treatment, and main group (64 patients), which received the standard treatment in combination with Immunovac VP4 vaccine..." This is not clear. What happen to the more severe and less severe groups? How were they re-assigned to the new "control" and "main" groups? In addition, the abstract reads "Blood monocyte PI correlates with COVID-19 severity:" This is just the opposite of what is reported later in the results section. As a consequence, the manuscript is disordered and confusing.

A major concern of the whole report is that the main purpose of the study is to evaluated the relationship between leukocyte phagocytic activity against S. aureus and clinical status. Unfortunately, phagocytosis is not evaluated properly. Since, phagocytic function is announced, from the title, as a major part of the study, one would expect a detailed description of the procedures to evaluate this important function. Phagocytosis assay is mentioned briefly and superficially. Many important technical details are left out. Because, phagocytosis was evaluated by flow cytometry, it is important to know how were cells identified? What cell markers were used? The cell gating strategy should be shown. Fluorescence associated to cells is not necessarily indicative of phagocytosis. How do authors distinguish internalized from adherent bacteria? Another confusing part is, how was phagocytosis estimated from "data tables corresponding to the flow cytometry histograms"? This is not clear at all, and it should be described in greater detail. More importantly, phagocytic index (PI) is not defined. Normally, phagocytic index is the number of phagocytosed particles (bacteria) per 100 phagocytic cells. In the present manuscript it is shown as "%". Percentage of what? In addition, it is mentioned that statistical methods were used to evaluate "kinetics of phagocytic indices". How could authors calculate kinetic parameters of phagocytosis from FACS data? Phagocytosis kinetics cannot be estimated with this methodology.

Another major flaw in this report is the lack of a proper control group. To accurately evaluate phagocytosis in cells from patients, phagocytosis of cells from healthy individuals "real control" must be included for comparison. Therefore, data presented do not support authors' conclusions and the whole report is not reliable.

In materials and methods section

Description of patients is cumbersome. Was the main group made up of patients with severe COVID? As indicated before the selection of groups is described in a confusing manner.

Description of Immunovac VP4 vaccine is also difficult to evaluate. Although, this product seems to be commonly used in Russia, there is very little literature about it to properly evaluate its properties. Articles related to this product and cited in the manuscript are in Russian. Therefore, it is not possible to really evaluate this product.

As indicated before, phagocytosis assay is not described properly.

In RESULTS section

Figure 1. Confusing. There is no legend. What are the red and white symbols?

Figure 2

Plots are not shown complete. They overlap. Also, the figure legend is confusing. Do all "more severe" and "less severe" patients "have moderate COVID? This is not a standard classification and it results very confusing.

For phagocytosis

"The median neutrophil PI was 97.9 (96.3; 99)%, and the median monocyte PI was 91.2 (84.6; 95)%. Phagocytic index is normally not expressed as percentage. It is not clear what data are been presented. What are the numbers in parenthesis? These data cannot be interpreted adequately. Moreover, more than 90 % does not suggest a problem with phagocytosis

Table 2

What is "RLMEM"?

Similarly, Rm and Rc are not defined

"Linear mixed-effects models (LMEM) were used to evaluate the relationship between the kinetics of phagocytic indices and the severity of a patient’s condition." Again, this statement is very confusing. Kinetics of phagocytosis cannot be evaluated with the methodology used in this report.

Figure 4

Again, what is monocyte phagocytic index? Data are not clear. No real difference is observed among groups. In all cases phagocytosis of cells from healthy "real control" individuals should be included for comparison.

In Discussion

"There is a growing body of evidence that phagocyte dysfunctions in patients with COVID-19 and lung damage are linked with an increased susceptibility to secondary infections." No references are provided for this statement.

"An upregulation of activation markers and downregulation of S. aureus phagocytosis in circulating phagocytes in COVID-19 patients was demonstrated by Koenis D.S. et al. [34, 35]"

The work by Koenis D.S. et al. is the reference 35. However, reference 34 does not support authors' statement. In fact, reference 34 suggests the opposite, as it reads "Fcγ receptor-mediated phagocytosis, IL17, and Tec kinase canonical pathways were enriched in patients with severe COVID-19".

Other points

Review the use of English language is required. Pronouns are missing in many sentences.

References 3 and 5 are the same

Reviewer #3: The primary aim of the manuscript "Phagocytic Activity of Blood Monocytes and Neutrophils in COVID-19 patients in relation to their clinical status and use of immunotherapeutic treatment" was to investigate the phagocytic function of neutrophils and monocytes in patients with moderate COVID-19 using the phagocytic index (PI). Specifically by analyzing changes in PI in patients treated with standard treatment and Immunovac VP4 compared to those receiving only standard treatment. PI was evaluated at different time points: immediately after hospitalization, and fourteen and thirty days post-hospitalization. The goal was to understand how phagocytic function differs between patients with more severe and less severe COVID-19 and how these changes correlate with other parameters such as C-reactive protein, oxygen saturation by pulse oximetry (SpO2), and aspartate aminotransferase (AST).

The main problem of the paper is the methodology, but other sections need to be reviewed.

1.- Clarification of Methodology:

- A complete description of how neutrophils and monocytes were purified from blood and how the phagocytic function was measured has to be included in the Methods section.

- Clearly understand how the phagocytic index (PI) was calculated.

2.- Consistency in Terminology:

- Standardizing the terminology throughout the manuscript is important for clarity. Using different terms "control," "main group," "standard treatment," and "standard treatment plus vaccine" to refer to the same group is very confusing. 

3.- Inclusion of Neutrophil Data:

- The abstract and results sections do not include data on the phagocytic index for neutrophils or address their phagocytic function alongside monocytes. Figures and text need to be updated accordingly.

4.- Figures and Data Presentation

- Figure 2 has to be revised to improve clarity and readability. The panels are incomplete.

- The graphs should better represent the data, highlighting the differences in PI between groups and time points.

- Figure legends should include a description of what colors represent on figures.

5.- Discussion and References:

- Ensure all paragraphs are appropriately referenced. The paragraph of the discussion section beginning with "considering a wide range of studies" does not include citations.

- The reference cited in the introduction regarding the ability of monocytes and macrophages to regenerate tissues is not an appropriate source.

6.- Title

- The title of the manuscript is misleading because it aims to analyze the phagocytic activity of blood monocytes and neutrophils in COVID-19 patients, and the methodology presented is not enough to drive the paper's conclusions.

Minor Changes:

7.- Abbreviations and Numbers:

- All abbreviations should be defined the first time they are mentioned in the manuscript.

- Numbers in parentheses, such as "CRP (20.1 (10.9; 43.2)) mg/L," need to be defined and clarified. Explain if these numbers represent medians and interquartile ranges.

8.- Abstract Adjustments:

- The abstract mentions monocytes as phagocytic cells but no neutrophils. The description of the results for neutrophils should be included.

9.- References:

- Duplicated references (e.g., references 3 and 5)

- References that do not support the corresponding paragraphs (e.g., reference 11)

10.- Additional edits:

- There are some typographical errors and inconsistencies in terminology.

- Define all abbreviations in tables and figures.

Reviewer #4: The authors of this brief research report investigate an important aspect of the relationship between the phagocytic activity of peripheral-blood monocytes and neutrophils and the severity of COVID-19. This study provides valuable insights into how the immune system responds to viral infections and how immune dysfunction may influence the course of the disease. It is noted that COVID-19 is often accompanied by bacterial infections, such as pneumonia.

The research presented is well-executed, with thorough attention given to the selection of study groups and parameters. However, there are a few major points for consideration.

Major Comments:

1. Novelty: the similar study was done by Laura Otto Walter et al. (2022, DOI:10.1111/imm.13457) with broader panel of biochemical parameters. Authors proposed as a novelty a study of Immunovac VP4 as an efficient treatment for COVID-19 accompanied infectious.

2. Monocyte PI: The statement that "higher monocyte PI on admission was associated with lower CRP levels, lower percentages of lung involvement on CT scan, and higher SpO2 levels" seems contradictory. This finding suggests a more complex interaction, possibly reflecting specific disease mechanisms. What could explain such effect?

3. Immunostimulant Effect: The treatment with the immunostimulant showed blood monocyte PI improvement only in more severe patients. What might be the underlying mechanism of Immunovac VP4 action in combination with standard treatment? Could standard treatment interfere with or inhibit the action of Immunovac VP4?

4. Duration of fever: “However, the more severe patients who received Immunovac VP4 had higher levels of oxygen saturation (p=0.01), which suggests clinical benefits of immunotherapy. Moreover, in the patients with less severe COVID-19 who received Immunovac VP4 the mean duration of fever was 2 days shorter than in those who received only the standard treatment (p=0.001).”

Quite curious results, which the authors probably should explain further in the Discussion section. Also, it’d be beneficial to include the duration of fever in more severe patients in the Results.

In the Results, the authors show that Immunovac VP4 did not affect the monocyte PI or SpO2, but it significantly reduced the duration of fever in less severe patients. Conversely, the opposite effects were observed in more severe patients.

Minor Comments:

1. The manuscript contains several typographical and grammar errors. For example, the phrase "monocyte and neutrophil phagocytic indices", “were investigated these parameters in relation to the treatment regimen administered to the study subjects.”

2. “In COVID-19, the activation of resident alveolar macrophages, CD68+CD169+ macrophages of lymph nodes, and CCR2+ monocytes through PRRs triggers the production of type I IFN-γ and other pro-inflammatory cytokines, such as interleukin (IL)-6, IL-1β, and tumor necrosis factor (TNF)-α, that have effective antiviral functions but also contribute to lung tissue damage and inflammation.” This statement requires a reference.

3. “In viral infections, neutrophils migrate to the site of pathogen invasion and contribute to the killing of viruses through active oxygen series (ROS) and phagocytosis.” Is it supposed to be Active oxygen species?

4. It’d be convenient to add more details in Figures description titles.

5. The manuscript needs language correction and Figure 2 rearrangement.

**Do you want your identity to be public for this peer review?** For information about this choice, including consent withdrawal, please see our Privacy Policy

Reviewer #1: No

Reviewer #2: No

Reviewer #3: No

Reviewer #4: No

---

## [Author Response · Author response to Decision Letter 1]

13 Apr 2025

Dear Editor! Dear Mrinmoy Sanyal!

We tried to change all your mentioned comments on the text! A correction of the original text of the manuscript, figures were carried out thanks to your valuable comments! We are very grateful for them!

Reviewer 1 - Thank you very much for your valuable comments and taking the time to read and study this article from the bottom of my heart! Dear and Respected Reviewer! First of all, thanks a lot for all your comments, valuable time spent for our manuscript reading and revision of the work! Your high rating is very important for us!!!

Reviewer 2 - Thank you very much for your time, attention and final comments that made our article better!

- Thanks to your comment, we have added a detailed description of the method:

“Phagocytic activity of monocytes and granulocytes (neutrophils) was assessed in heparinized peripheral blood samples by measuring leukocyte phagocytic absorption activity against heat-killed S. aureus labeled with fluorescein isothiocyanate. Blood samples were analyzed by flow cytofluorometry on an FC-500 BeckmanCoulter flow cytometer. Daily cultures, the second passage of S. aureus Wood 46, were washed with isotonic sodium chloride solution, killed by heating to 96-98 ° C for 40 minutes, precipitated at 1000 g for 25 minutes, washed twice in 10 ml of phosphate-buffered saline (PBS), pH 7.4. According to the turbidity standard, the concentration of bacteria was brought to 200 million/ml with carbonate-bicarbonate buffer pH 9.5. FITC (Sigma) was added to the bacterial suspension at a final concentration of 0.1 mg/ml and incubated at +4°C for 12 hours. Then unbound FITC was removed by washing three times PBS solution on 1000 g for 25 minutes. The bacterial suspension was aliquoted and stored at +4°C for 1 month, at -70°C up to 6 months. Whole heparinized blood was used to set up the reaction. A suspension of FITC-labeled staphylococci and blood cells in a ratio of 1:10 was placed in Eppendorf tubes and incubated at 37°C for 30 minutes. Quenching of FITC-labeled bacteria adhered to the surface of leukocytes was performed by adding a solution of trypan blue (0.2 mg/ml). Then Optilyse C solution was added to lyse erythrocytes and the samples were incubated at room temperature in the dark for 30 minutes. Next, cold phosphate-buffered saline, pH 7.2-7.4, with 0.02% EDTA was added to stop the phagocytic reaction. After 3-fold washing with ISOTON II solution, the samples were analyzed on a Beckman Coulter FC-500 flow laser cytometer. The cytometer settings were set so that three cell clouds - granulocytes (neutrophils), monocytes, and lymphocytes - were conveniently placed on the forward scatter (FSC) and side scatter (SSC) diagram in the Dot Plot window. Phagocytic cells were counted based on the intensity of green fluorescence (FL-1 channel). The percentage of phagocytic granulocytes and monocytes in the test sample was based on the cytometry results. The recommended number of events collected for neutrophils is 3000.”

We also edited main concepts following the text: instead of “phagocytic indices” it became “percentage (%) of phagocytic leukocytes (neutrophils and monocytes)”.

- Thank you for your comment! Unfortunately, the personal translator has very inaccurately reflected the essence of what was written in the original language. Of course, "kinetics of phagocytic indices" cannot be measured, but in the original version it was "dynamics of indicators". It is our fault, thank you for making our article better!

- At the height of the COVID-19 pandemic, when there was no specific vaccine against SARS-CoV-2 worldwide, various methods were taken to save the lives of people. In addition, the introduction of a self-isolation regime for all residents in the country prohibited the opportunity to move around the city, so examination of healthy people was not possible at that time. The fact that healthy people were not admitted to hospitals also complicates the solution of this problem, all practical health care resources worked in the overwhelming majority to save the lives of patients with COVID-19, and among the medical staff of hospitals (as the only “available” healthy people as a control group) it was difficult to find those not yet infected.

- Now it is described in a more exact way:

“The study population was comprised of 69 males and 36 females. The mean age of the patients was 43.5 (37; 51) years old; the mean body mass index 27.4 (25; 30.1) kg/m2; and the mean duration of disease prior to hospitalization 6 (4; 8) days. According to the officially recommended severity criteria (T body > 38 °C; RR > 22/min, dyspnea during physical exertion; changes in CT (radiography) typical for viral infection; SpO2 < 95%; CRP in serum > 10 mg/l.) patients with moderate course of the disease were divided into two groups: less severe and more severe. This was done to see the dependence of changes in phagocytic activity on the severity of the disease. Since disease severity depends on a number of factors, a cluster analysis was carried out to classify all the patients (n=105) into clusters based on several parameters (Table 1).

A comparison was also made between patients (n=105) with moderate COVID-19 depending on the type of treatment received. The control group (group with standard treatment) consisted of 41 patients with moderate COVID-19 (27 males and 14 females, median age 42 [33-54]) who received only the standard treatment. The main group (group with standard treatment plus vaccine) was made up of 64 patients (42 males and 22 females, median age 42 [37-45]) who received the standard treatment accompanied by Immunovac VP4 vaccine, a bacteria-based immunostimulating agent, starting from day 1 of hospitalization.

The main and control groups were matched by age (p=0.79), gender (p=0.33) and the number of days between the onset of disease and hospitalization (5 [4-8] days in Group 1 and 5 [3-7] days in Group 2, p=0,63). The patients in both groups were also matched by body mass index, amount of impaired lung parenchyma, and laboratory findings”.

In other words: Cluster analysis was used for the entire cohort of patients with COVID (n=105), regardless of the therapy administered, since the initial data of patients at the time of admission to the hospital were used. According to the officially recommended severity criteria (T body > 38 °C; RR > 22/min, Dyspnea during physical exertion; Changes in CT (radiography) typical of viral infection; - SpO2 < 95%; - CRP in serum > 10 mg / l.) patients with moderate course of the disease were divided into two groups: less severe and more severe. This was done to see the dependence of changes in phagocytic activity on the severity of the disease.

Main group: a group of patients only with moderate course of the disease, who in addition to standard therapy were given Immunovac -VP4.

Control: a group of patients with moderate course of the disease, who received only standard therapy.

- "Immunovac VP4" belongs to the class of therapeutic vaccines of bacterial origin. Analogues of "Immunovac VP4" are Bronchomunal, Bronchovaxom and Respivax, consisting of lysates of 7 types of opportunistic bacteria, intended for oral use only. Immunovac VP4 has a number of advantages: it contains antigens of 4 types of specially selected highly immunogenic bacterial strains with broad intraspecific and interspecific cross-protective activity, including against various serotypes of Streptococcus pneumoniae and Hemophilus influenzae. To obtain the antigens included in "Immunovac VP4", gentle methods of action on bacterial cells were selected, allowing to preserve the native structure of antigens. The course of therapy with Immunovac VP4, administered orally or subcutaneously, is 1 month, the duration of the therapeutic effect is a year or more, while the course of treatment with Broncho-munal and its analogues is 3 months with the possibility of repeat after 6 months.

Moreover, not so long ago an article was published in the journal Nature on the use of this bacterial lysate in the treatment and improvement of outcomes of COVID-19.

1) Kostinov, M., Svitich, O., Chuchalin, A. et al. Secretory IgA and course of COVID-19 in patients receiving a bacteria-based immunostimulant agent in addition to background therapy. Sci Rep. 2024. 14: 11101. https://doi.org/10.1038/s41598-024-61341-7

2) De Benedetto F., Sevieri G. Prevention of respiratory tract infections with bacterial lysate OM-85 bronchomunal in children and adults: a state of the art. Multidiscip Respir Med [Internet]. 2013 May 22 [cited 2025 Mar. 2];8(9). Available from: https://mrmjournal.org/index.php/mrm/article/view/510

3) Choi J.Y., Park Y.B., An T.J. et al. Effect of Broncho-Vaxom (OM-85) on the frequency of chronic obstructive pulmonary disease (COPD) exacerbations. BMC Pulm Med 23, 378 (2023). https://doi.org/10.1186/s12890-023-02665-4

4) Nikolova M., Stankulova D., Taskov H., et al. Polybacterial immunomodulator Respivax restores the inductive function of innate immunity in patients with recurrent respiratory infections. International Immunopharmacology. 2009, 9(4):425-432.

DOI:10.1016/j.intimp.2009.01.004

- Figure 1 has been changed. It seems to us that it is more visual this way. The legend has been added. The red and white symbols – “Cluster 1” and “Cluster 2”.

- We made graphs like pictures so that they wouldn’t overlap. Renamed them "Cluster 1" and "Cluster 2". If necessary, we can send all the drawings as separate files (jpeg format, high quality - dpi 1200). Additionally, clearer descriptions of the groups "more severe" and "less severe" patients with precise characteristics have been added throughout the text (in different sections).

- Throughout the text, the term phagocytic index has been changed to the percentage (%) of phagocytic monocytes and/or neutrophils.

- RLMEM - robust linear mixed-effects models.

Rm - marginal R squared (only fixed effects), Rc - conditional R-squared (fixed and random effects).

- It has been added to the text thanks to your comment! Unfortunately, the personal translator has very inaccurately reflected the essence of what was written in the original language. Of course, "kinetics of phagocytic indices" cannot be measured, but in the original version it was "dynamics of indicators". It is our fault, thank you for making our article better!

- Throughout the text, the term phagocytic index has been changed to the percentage of phagocytic monocytes and neutrophils.

- References have been added and updated.

- The revised version of the manuscript contains a modified list of references due to the corrections made.

- References have been updated. Text has been edited.

Reviewer 3 - Thanks a lot for your time and attention paid for our manuscript! Undoubtedly, all your valuable comments are taken into account!

- Thanks to your comment, we have added a detailed description of the method:

“Phagocytic activity of monocytes and granulocytes (neutrophils) was assessed in peripheral blood samples by measuring leukocyte absorption activity against heat-killed S. aureus labeled with fluorescein isothiocyanate. Blood samples were analyzed by flow cytofluorometry on an FC-500 BeckmanCoulter flow cytometer. Daily cultures, the second passage of S. aureus Wood 46, were washed with isotonic sodium chloride solution, killed by heating to 96-98 ° C for 40 minutes, precipitated at 1000 g for 25 minutes, washed twice in 10 ml of phosphate-buffered saline (PBS), pH 7.4. According to the turbidity standard, the concentration of bacteria was brought to 200 million/ml with carbonate-bicarbonate buffer pH 9.5. FITC (Sigma) was added to the bacterial suspension at a final concentration of 0.1 mg/ml and incubated at +4°C for 12 hours. Then unbound FITC was removed by washing three times PBS solution on 1000 g for 25 minutes. The bacterial suspension was aliquoted and stored at +4°C for 1 month, at -70°C up to 6 months. Whole heparinized blood was used to set up the reaction. A suspension of FITC-labeled staphylococci and blood cells in a ratio of 1:10 was placed in Eppendorf tubes and incubated at 37°C for 30 minutes. Quenching of FITC-labeled bacteria adhered to the surface of leukocytes was performed by adding a solution of trypan blue (0.2 mg/ml). Then Optilyse C solution was added to lyse erythrocytes and the samples were incubated at room temperature in the dark for 30 minutes. Next, cold phosphate-buffered saline, pH 7.2-7.4, with 0.02% EDTA was added to stop the phagocytic reaction. After 3-fold washing with ISOTON II solution, the samples were analyzed on a Beckman Coulter FC-500 flow laser cytometer. The cytometer settings were set so that three cell clouds - granulocytes (neutrophils), monocytes, and lymphocytes - were conveniently placed on the forward scatter (FSC) and side scatter (SSC) diagram in the Dot Plot window. Phagocytic cells were counted based on the intensity of green fluorescence (FL-1 channel). The percentage of phagocytic granulocytes and monocytes in the test sample was based on the cytometry results. The recommended number of events collected for neutrophils is 3000.”

We also edited main concepts following the text: instead of “phagocytic indices” it became “percentage (%) of phagocytic leukocytes (neutrophils and monocytes)”.

- Clearer descriptions of the groups "more severe" and "less severe" patients, "control," "main group," "standard treatment," and "standard treatment plus vaccine" with precise characteristics have been added throughout the text (in different sections).

Now it is described in a more exact way:

“The study population was comprised of 69 males and 36 females. The mean age of the patients was 43.5 (37; 51) years old; the mean body mass index 27.4 (25; 30.1) kg/m2; and the mean duration of disease prior to hospitalization 6 (4; 8) days. According to the officially recommended severity criteria (T body > 38 °C; RR > 22/min, dyspnea during physical exertion; changes in CT (radiography) typical for viral infection; SpO2 < 95%; CRP in serum > 10 mg/l.) patients with moderate course of the disease were divided into two groups: less severe and more severe. This was done to see the dependence of changes in phagocytic activity on the severity of the disease. Since disease severity depends on a number of factors, a cluster analysis was carried out to classify all the patients (n=105) into clusters based on several parameters (Table 1).

A comparison was also made between patients (n=105) with moderate COVID-19 depending on the type of treatment received. The control group (group with standard treatment) consisted of 41 patients with moderate COVID-19 (27 males and 14 females, median age 42 [33-54]) who received only the standard treatment. The main group (group with standard treatment plus vaccine) was made up of 64 patients (42 males and 22 females, median age 42 [37-45]) who received the standard treatment accompanied by Immunovac VP4 vaccine, a bacteria-based immunostimulating agent, starting from day 1 of hospitalization.

The main and control groups were matched by age (p=0.79), gender (p=0.33) and the number of days between the onset of disease and hospitalization (5 [4-8] days in Group 1 and 5 [3-7] days in Group 2, p=0,63). The patients in both groups were also matched by body mass index, amount of impaired lung parenchyma, and laboratory findings”.

In other words: Cluster analysis was used for the entire cohort of patients with COVID (n=105), regardless of the therapy administered, since the initial data of patients at the time of admission to the hospital were used. According to the officially recommended severity criteria (T body > 38 °C; RR > 22/min, Dyspnea during physical exertion; Changes in CT (radiography) typical of viral infection; - SpO2 < 95%; - CRP in serum > 10 mg / l.) patients with moderate course of the disease were divided into two groups: less severe and more severe. This was done to see the dependence of changes in phagocytic activity on the severity of the disease.

Main group: a group of patients only with moderate course of the disease, who in addition to standard therapy were given Immunovac -VP4.

Control: a group of patients with moderate course of the disease, who received only standard therapy.

- Figure 1 has been changed. It seems

---

## [Decision Letter · Decision Letter 1]

5 May 2025

Phagocytic activity of blood monocytes and neutrophils in moderate COVID-19 patients and impact of immune therapy with bacterial lysates

PONE-D-24-46558R1

Dear Dr. Kostinov,

We’re pleased to inform you that your manuscript has been judged scientifically suitable for publication and will be formally accepted for publication once it meets all outstanding technical requirements. Thank you for the valuable contribution.

Kind regards,

Mrinmoy Sanyal, PhD

Academic Editor

PLOS ONE

Reviewers' comments:

Reviewer's Responses to Questions

**Comments to the Author**

Reviewer #1: All comments have been addressed

Reviewer #4: All comments have been addressed

2. Is the manuscript technically sound, and do the data support the conclusions?

Reviewer #1: Yes

Reviewer #4: Yes

3. Has the statistical analysis been performed appropriately and rigorously?

Reviewer #1: Yes

Reviewer #4: Yes

4. Have the authors made all data underlying the findings in their manuscript fully available?

Reviewer #1: Yes

Reviewer #4: Yes

5. Is the manuscript presented in an intelligible fashion and written in standard English?

Reviewer #1: Yes

Reviewer #4: Yes

Reviewer #1: Very Respected Authors,

The authors have made all the corrections. The abstract is well writen. The objective is clear. The method has been updated. The results are presented more clearly.

Reviewer #4: (No Response)

**Do you want your identity to be public for this peer review?** For information about this choice, including consent withdrawal, please see our Privacy Policy

Reviewer #1: No

Reviewer #4: No

---

## [Editor Report · Acceptance letter]

PONE-D-24-46558R1

PLOS ONE

Dear Dr. Kostinov,

I'm pleased to inform you that your manuscript has been deemed suitable for publication in PLOS ONE. Congratulations! Your manuscript is now being handed over to our production team.

Kind regards,

on behalf of

Dr. Mrinmoy Sanyal

Academic Editor

PLOS ONE